

**Comparison of retrieved Noctilucent cloud particle**
**properties from Odin tomography scans and model**
**simulations**
**Linda Megner[1] , Ole M. Christensen[1], Bodil Karlsson[1], Susanne Benze[1], and**
**Victor I. Fomichev[2]**
[1]{Department of Meteorology, Stockholm University, Sweden}
[2]{CRESS, York University, Canada}
Correspondence to: L. Megner (linda@misu.su.se)
**Abstract**
Mesospheric ice particles, known as Noctilucent clouds or Polar Mesospheric Clouds, have
long been observed by rocket instruments and satellites, while models have been used to
simulate ice particle growth and cloud properties. However, the fact that different
measurement techniques are sensitive to different parts of the ice particle distribution makes it
difficult to compare retrieved parameters such as ice particle radius or particle number density
from different experiments. In this work we investigate the accuracy of satellite retrieval
based on scattered light and how this affects derived cloud properties. We run the retrieval
algorithm on modelled cloud distributions and compare the results to the properties of the
original distributions. We find that ice mass density is accurately retrieved whereas mean
radius often is overestimated and high number densities generally are underestimated. The
reason is that the retrieval algorithm assumes a Gaussian size distribution, whereas the
modelled size distributions often are multimodal. Once we know the limits of the satellite
retrieval we proceed to compare the properties retrieved from the modelled cloud distributions
to those observed by the Optical, Spectroscopic, and Infrared Remote Imaging System
(OSIRIS) instrument on the Odin satellite. We find that a model with a stationary atmosphere,
as given by average atmospheric conditions, does not yield cloud properties that are in
agreement with the observations, whereas a model with realistic temperature and vertical





wind variations does. This indicates that average atmospheric conditions are insufficient to
understand the process of Noctilucent cloud growth and that a realistic atmospheric variability
is crucial for cloud formation and growth. Further, the agreement between results from the
model - when set up with a realistically variable atmosphere - and the observations suggests
that our understanding of the growth process itself is reasonable.

## 7   1   Introduction

At the summer polar mesopause, the coldest region on Earth, the temperature drops low
enough so that ice particles can form despite the low water content of a few parts per million.
These ice clouds, known as Noctilucent clouds (NLCs) or Polar Mesospheric Clouds (PMCs),
provide a way to monitor this remote region of the atmosphere, where in situ measurements
can only be carried out using rockets. NLCs have been observed by the naked eye since the
late 19[th] century (Leslie, 1885) and since the second half of the 20[th] century, rocket
instruments, satellites, lidars and models have been used to develop our understanding of the
clouds (e.g. Witt, 1960;Turco et al., 1982;Barth et al., 1983;Hansen et al., 1989).
The different measurement techniques used in remote sensing and for in situ measurements -
and even by particular types of instruments within these categories - make it difficult to
compare retrieved parameters such as ice particle radius or particle number density from
different experiments. For example, many in situ rocket measurements are not sensitive to the
size of the particles, as long as they are above a certain aero-dynamical threshold that is
determined by the shape of the instrument and the speed of the rocket (Hedin et al., 2007).
Remote sensing instruments like satellites and lidars on the other hand, are more sensitive to
the particles that more efficiently scatter or absorb light, i.e. the particles at the larger end of
the size distribution. They, in particular the instruments that observe scattered light, are thus
rather insensitive to the smaller end of the size distribution. A direct comparison of for
example the number densities measured by in situ and remote sensing techniques is therefore
not straight forward.
Even comparisons between individual satellite observations have proven very difficult (Bailey
et al., 2015). These difficulties are also due to the fact that different measurement techniques
inevitably favour different parts of the size distribution. For instance, an instrument that
measures the absorption of light will be sensitive to the total volume of the ice while an
instrument that observes scattered light will be sensitive to different regions of the size





distributions depending on what scattering angles it observes. If, as earlier studies have
indicated, the size distribution were truly Gaussian with a certain width (see e.g. Rapp and
Thomas, 2006), then this problem would be easier to overcome, but as will be shown in this
study, our model simulations suggest that this is not generally the case.
The size distribution of ice particles in the cloud layer varies with altitude (models predict that
they range from hundreds or thousands freshly nucleated small particles per cubic centimetre
at the mesopause to ten or less more mature particles per cubic centimetre at approximately
81-83 km; (Megner, 2011). This means that the question of which part of the size distribution
an instrument is sensitive to is intricately connected to which altitude region the instrument is
sensitive to.
In this paper we therefore first investigate the accuracy of the Odin satellite's retrieval of
properties such as ice water content (IWC), mean radius, and total number density. We do this
by running the retrieval algorithm on modelled cloud distributions (which obviously are fully
known) and comparing the retrieved results to the properties of the original distributions.
After this we proceed to compare the properties retrieved from the modelled cloud
distributions to those observed by satellite. We use satellite observations from the Odin
tomography modes (Hultgren et al., 2013) for which the satellite's scanning sequence is
specifically designed to provide multiple measurements through the same cloud volume,
which enables, via tomography, high resolution altitude and horizontal observations of the
NLCs. We use information from both instruments on-board the Odin satellite: the Optical
Spectrograph and InfraRed Imager System (OSIRIS) instrument (Llewellyn et al., 2004) gives
us high resolution data of the NLCs and the Sub-Millimeter Radiometer (SMR) instrument
(Nordh et al., 2003) provides information of the background temperature and water vapour,
which in this experiment are used as input to our model.
The specific aims of this study are to:
1) Identify what part of the size distribution we capture with an OSIRIS-type measurement
and to evaluate to what extent retrieved properties - such as mean radius, IWC and particle
number density - of the sampled volume represent corresponding actual properties.
2) Investigate if our current knowledge of the microphysics (as represented by the CARMA-
model) is accurate enough to simulate clouds that match our observations, and to pinpoint
what model input is crucial for simulating representative clouds.





The paper is structured as follows: In section 2 the Odin tomography scans and the retrieval
algorithms of OSIRIS and SMR are described. In section 3 the microphysical model is
described. Section 4 gives the results of the comparisons and finally section 5 summarizes the
conclusions.
**2   Odin tomography scans**
Both OSIRIS and SMR observe the atmosphere in the limb geometry: the co-aligned optical
axes of both instruments sweep over a selected altitude range in the forward direction as the
entire satellite is nodded up and down. During the stratosphere/mesospheric mode, both
instruments scan from 7 to 107 km. However, during the tomography mode, only the NLC
region of interest, 78 to 90 km, is scanned. This decreases the horizontal distance between
subsequent scans and increases the number of lines of sight through a given atmospheric
volume, thus enabling the tomographic retrieval of cloud and background atmosphere
properties. During the NH10 and NH11 seasons, a total of 180 orbits were performed using
the tomographic mode. The orbits were chosen to provide coincident observations with the
Aeronomy of Ice in the Mesosphere (AIM) satellite and cover three three-day periods during
each NLC season (Table 1, Hultgren et al., 2013). A tomographic retrieval algorithm is then
used to convert the limb-integrated atmospheric line-of-sight properties into local information
about cloud properties or the background atmosphere (Christensen et al., 2015;Hultgren and
Gumbel, 2014;Hultgren et al., 2013). Using the tomographic algorithm these local properties
can be retrieved between 78 and 87 km with a horizontal and vertical resolution of ~330 km
and 1 km, respectively. For this analysis we use four days of tomographic data (76 scans)
between 70°N and 77 °N of July 2010 and 2011, where SMR and OSIRIS data both are
available. During these days, clouds and background atmosphere were sampled at Solar
Scattering Angles of 70° to 100°.
**2.1  OSIRIS retrieval**
The tomographic algorithm transforms the observed OSIRIS limb radiances into the retrieved
volume scatter coefficient, a measure of cloud brightness. In contrast to the input limb
radiance, which is dependent on tangent altitude and thus contains signals from fore- and
background, the retrieved volume scatter coefficient is a local signal dependent on the vertical





dimension altitude and the horizontal dimension Angle Along Orbit (AAO). The algorithm
used is the Multiplicative Algebraic Reconstruction Technique (MART) based on maximum
probability techniques (Hultgren et al., 2013; 2014).
OSIRIS observes scattered sunlight at wavelengths between 277 and 810 nm, with a
resolution of approximately 1 nm. For this study, the volume scatter coefficient at specific
wavelengths in the UV-range (277.3 nm, 283.5 nm, 287.8 nm, 291.2 nm, 294.4 nm, 300.2 nm,
and 304.3 nm; see e.g. Karlsson and Gumbel, 2005, for details) is used to retrieve particle
sizes from the OSIRIS radiance measurements by fitting the observed spectral signal to
tabulated scattering spectra from numerical T-matrix simulations (Baumgarten and Fiedler,
2008; Mishchenko and Travis, 1998). Once a particle mode radius is retrieved, number
density, and ice mass density can be estimated. The retrieval assumes a Gaussian particle size
distribution, with a width that varies as 0.39 times the retrieved mean radius but stays fixed at
15.8 nm for larger radii (Baumgarten et al., 2010). Further, the particles are assumed to be
oblate spheroids with and axial ratio of 2. The retrieval size for mode radius is constrained to
< 100 nm. This is because it is not possible to distinguish between particles > 100 and smaller
particles (around 50 nm) in the approach we are using (see e.g. von Savigny and Borrows,
2007, for an equivalent issue). A consequence of this constraint is that the algorithm will
select a small mode radius that fits the signal even in the presence of really large particles.
Whether this is an acceptable shortcoming in the retrieval algorithm or not is out of the scope
of this study; our conclusions are not affected by this constraint.
The PMC microphysical retrieval and resulting uncertainties in cloud brightness and
microphysical products are described in detail by Hultgren et al. (2013) and Hultgren and
Gumbel (2014). Based on uncertainty in the input radiances, they estimate a typical statistical
error in cloud brightness of $10^{-11}$ m$^{-1}$ str$^{-1}$, which is less than 1% of the typical NLC peak
brightness. Propagating the error of the individual radiances through the tomographic retrieval
algorithm, statistical uncertainties in mode radius (~ ±6 nm throughout all altitudes), number
density (from ±1 cm$^{-3}$ at 81 km to ±35 cm$^{-3}$ at 86 km), and ice mass density (negligible at
lower PMC altitudes, up to ±5 ng m$^{-3}$ at 86 km) are estimated.
**2.2   SMR retrieval**
SMR measures thermal emission from the 557 GHz water vapour line. From this, the
concentration of water vapour and temperature can be retrieved in the aforementioned altitude



region. This can be achieved as the line is very strong and becomes optically thick even in the
MLT region. The retrieval is done using the non-linear optimal estimation method with a
Levenberg-Marquardt iteration scheme. The resulting precision is 0.2 ppmv for water vapour
mixing ratio and 2 K for temperature. The data used in this study are all collected when SMR
was operating in frequency mode 13, as this mode shows the best agreement with other
satellite instruments (within 5 K for temperature and 20% for water vapour). For further
details see Christensen et al. (2015).
**3   CARMA model**
Community Aerosol and Radiation Model for Atmospheres (CARMA) is a microphysical
cloud    model    that    originated    from    a    stratospheric    aerosol    code
(Toon et al., 1979;Turco et al., 1979) that was developed to simulate clouds in a variety of
environments ranging from the Earth's atmosphere to other planetary atmospheres. It has
been used to simulate NLCs in numerous publications (e.g. Asmus et al., 2015;Chandran et
al., 2012;Megner, 2011;Megner et al., 2006;Rapp and Thomas, 2006;Merkel et al.,
2009;Stevens, 2005;Vergados and Shepherd, 2009;Lübken et al., 2007). As in the majority of
these studies, we use the 1-dimensional setup of the model to simulate microphysical
processes such as ice nucleation and growth, sedimentation and vertical transport. Three
interactive constituents are simulated: Condensation Nuclei (CN), ice particles and water
vapour. The CN are assumed to be meteoric smoke particles with a density of 2 $g/cm^3$. The
number density and size distribution of the CN are representative of the middle of the NLC
season (July 10th) at 68°N (see Figure 1 in Megner et al. (2008a). The nucleation is treated in
the framework of droplet theory (Fletcher, 1958) where the probability of nucleation depends
on the size of the CN and the contact angle. The contact angle, also known as the wettability,
in turn depends on the surface energies between nucleus, ice and air (Fletcher, 1958;Keesee,
1989;Gumbel and Megner, 2009;Megner and Gumbel, 2009). While this quantity remains
uncertain, it has been argued that meteoric smoke acts very efficiently as ice nuclei (Roddy,
1984;Rapp and Thomas, 2006) and the contact angle is therefore set to 0.95 in agreement with
previous studies (Megner, 2011;Megner et al., 2008a;Rapp and Thomas, 2006). The model
domain spans from 72 to 102 km in altitude with a resolution of 0.25 km. The ice particles are
considered spherical and the size distributions are evaluated on radius grids consisting of 40
non-equally spaced size bins between 2 to 900 nm. The piecewise parabolic method algorithm
(Colella and Woodward, 1984) is used for both vertical advection and deposition growth





(advection in particle radius space) with a time step of 100 s. Following Rapp and Thomas
(2006) we use an eddy diffusion profile adapted from the collection of turbulence
measurements at 69° N under polar summer conditions (Lübken, 1997).

## 5   4   Results

As explained in Section 2, the Odin tomography scans give us simultaneous high resolution
observations of ice particles from OSIRIS and water vapour and temperature from SMR. We
use these SMR observations as input to the CARMA model and then compare the modelled
clouds to those observed by OSIRIS. However, we cannot use the water vapour and
temperature profiles from an SMR observation that is made simultaneously to the OSIRIS
observation of ice particle properties as initial state for the model. The reason is that ice
growth is not an instantaneous process, i.e. the environment that the clouds grow in is not
necessarily the same as the environment they are observed in. For instance the ice growth
process itself uses up much of the available water, leaving a depleted water profile. Since we
do not have any observations of the history of the atmospheric environment in which the
cloud developed we cannot compare a single observed cloud directly to its modelled
equivalent. We therefore have to settle for a more statistical approach, by comparing general
clouds that are observed by OSIRIS to modelled clouds that have developed in the typical
atmospheric environment that SMR observes. In Sections 4.1 and 4.2 we investigate two ways
of creating such a typical environment from the SMR observations and report about the
clouds they produce. As presented in the introduction, one main goal of this study is to
identify what part of the size distribution we capture with the OSIRIS-type instrument
retrieval and how this is reflected in the retrieved properties such as mean radius, IWC and
number density of particles. In Section 4.3 we investigate this by running the retrieval
algorithm on modelled cloud distributions and comparing the retrieved results to the original
distribution. Finally, in Section 4.4 we compare the modelled clouds to those observed by
OSIRIS.

## 28   4.1   The Stationary Atmosphere

In order to generate a typical cloud growth environment from the SMR measurements we
select observations that are co-located with the OSIRIS tomography scans where no clouds
were present. By selecting only the measurements where no clouds are present we avoid the





problem of not accounting for water that is already in the ice phase. We then calculate the
average water vapour and temperature profiles and use these fields to drive the model. Since
SMR data is only trustworthy up to an altitude of 87 km we extended the water vapour profile
linearly above this altitude, while for the temperature profile we used the SABER profile from
Sheese et al. (2011) as shown in Figure 1. Since SMR does not measure vertical wind we
follow Rapp and Thomas (2006) and a vertical wind profile representative of 69N as given by
Berger (2002). The temperature, water vapour and wind profiles in this run are thus
stationary. In this model setup, only a very minor IWC of maximum 0.03 ng/m$^3$ developed.
This is far below the detection threshold of OSIRIS of 5 ng/m$^3$. Hence, if the model is driven
by mean atmospheric conditions as measured by the SMR instrument it will not produce
visible clouds. The main reason is simply that the small (fraction of 1 nm) meteoric smoke
particles are not efficient condensation nuclei at a temperature of approximately 131 K (the
mesopause temperature shown in Figure 1), see Gumbel and Megner (2009). We note that
the model setup used in Rapp and Thomas (2006) does in fact result in observable clouds.
This is because they use the meteoric smoke distribution of Hunten et al. (1980), which is
based on a one-dimensional model of ablation and recombination of meteoric material and as
such lacks meridional atmospheric transport. More recently multi-dimensional models have
shown that this transport efficiently depletes the summer mesopause of meteoric material
resulting in much smaller meteoric smoke particles in this region than what was earlier
assumed (Megner et al. 2008b, Bardeen et al. 2008).
The SMR average temperature is declining with altitude up to 87 km, where the measurement
quality is diminishing. Thus, it gives no information on where exactly the mesopause is. To
examine if a higher (and thus colder) mesopause would trigger the model to produce clouds,
the temperature profile above the SMR observations was extended to lower temperatures and
a higher mesopause using the OSIRIS temperatures (Sheese et al., 2011) as shown by the
dash-dotted line in Figure 1. Although this resulted in a larger IWC of maximum 2 ng/m$^3$, it is
still below the detection threshold of OSIRIS.
In order to investigate how much colder the atmosphere needs to be for the model to produce
clouds, the average temperature profile was reduced in steps of 1K, and used as input to the
model. In order to produce clouds in CARMA of similar IWC as the clouds observed by
OSIRIS, the temperature profile had to be reduced by 6 K. However the particles produced by
this model realization were too large (150 nm) and their number densities far too small (<10



particles/cm$^3$ throughout the cloud region) compared to observations. Apparently, clouds from
this model run were not a realistic representation of the clouds we observe with Odin. We can
conclude that a simple shift of the temperature profile towards lower values is not enough to
produce realistic NLCs.
Another possibility to facilitate cloud formation is to assume that the CNs are larger, or more
efficient, so that they can nucleate ice particles at a higher temperature. To test this we first
enhanced the contact angle to unity, i.e. perfect wettability (see Section 3). This did not have a
major effect on the cloud properties and resulted in a maximum ice water density of 0.4
ng/m$^3$, which is still far below the OSIRIS detection limit. However, the CN distribution is
dependent on many uncertain parameters (Megner et al., 2006). For instance, if there is more
meteoric influx into the atmosphere, if the CNs are electrically charged (Gumbel and Megner,
2009;Megner and Gumbel, 2009), or if there is more coagulation within the meteor trail than
what is generally assumed in models of meteoric coagulation and transport (Megner et al.,
2008b;Bardeen et al., 2008), then this could result in a CN distribution that is more efficient
for nucleation. Thus we pose the question: What is the number density of efficient CNs
required to generate clouds with an IWC that agree with the OSIRIS observations? To answer
this question we assumed simple mono-sized distributions of particles with radii of 2 nm, i.e.
large enough to be efficient CN at 131 K  (Gumbel and Megner, 2009) but small enough not
to rapidly sediment out of the mesopause region. Note, that for simplicity we here enhance the
condensation nuclei efficiency by making the particle larger, but the nucleation efficiency can
be enhanced by other means, such as charging of the particles, with equivalent results. By
feeding the model mono-sized particle distributions of 10, 100, 1000 and 10000 particles/cm$^3$
we determined that approximately 100 efficient CNs /cm$^3$ was needed to produce an ice mass
equivalent to the OSIRIS observations. It should be noted that increasing the number of CNs
even more has little effect on the ice mass, as pointed out by Megner (2011); the case with
10000 particles/cm$^3$ gave approximately twice the ice mass compared to the case with 100
particles/cm$^3$. Despite that a CN distribution consisting of 100 particles/cm$^3$ of 2 nm radii is
not considered likely  - the original CN distribution from the model by Megner (2011) falls
sharply with radius and has on the order of 10 particles larger than 1 nm and 10$^{-4}$ particles/cm$^3$
larger than 2 nm - we nevertheless show the cloud generated in this way in Figure 2, as an
example of a cloud generated in stationary conditions with a highly efficient CN distribution.
This cloud will be referred to as the "No Wave" cloud.



It is however clear that the most straight forward solution to the lack of cloud development in
an averaged steady state atmosphere is not that a more efficient size distribution is needed, but
simply that the ice particles observed in the real atmosphere are nucleated during the times
when the temperature is below the average. This we will investigate in the next section.

## 4.2   Variable atmosphere

The mesopause region is characterized by high wave activity (e.g. McLandress et al., 2006).
This means that the constant temperature profile achieved by averaging the SMR
measurements as describe above is not representative. In order to represent the fast
temperature variations and vertical winds that give rise to them, we use July temperature and
vertical wind fields from July 69°N from the extended Canadian Middle Atmosphere model
(CMAM) (Beagley et al., 2010;Fomichev et al., 2002;McLandress et al., 2006) with a high
temporal resolution output (30 minutes). In this second setup of the CARMA model we still
use the SMR retrieved mean temperature profile to determine the average conditions, but
impose the time resolved CMAM temperature field to represent the temperature variations. In
practice this is achieved by adding a constant temperature shift to the CMAM data so that the
average CMAM temperature profile matches up with the average measured SMR profile. The
resulting temperature profile, and the associated temperature variations are shown in Figure
3a and b. As can be seen the variations from the CMAM model are fairly similar to those of
the SMR data set, especially given that the CMAM variations include diurnal variations
which are not well sampled by SMR since SMR measures predominantly at two local times.
The variations of the CMAM model also agree well with observations of daily variations in
the summer polar mesopause region (Höffner and Lübken, 2007). Since the vertical wind is
intimately connected to the temperature via adiabatic heating/cooling, we use the
accompanying CMAM vertical wind field to drive our model simulations (Figure 3 c and d).
The output from the CMAM model was fed into CARMA at time steps of 30 minutes.
This second model setup, which includes variations in temperature and winds, resulted in
clouds of IWC above the OSIRIS detection threshold and, as we shall see, of similar IWC as
that measured by OSIRIS. An example of a cloud produced in this way can be seen in the
lower panel of Figure 2. We will refer to these clouds as "Wave" clouds.





### 4.3  Modelled cloud retrieval

An important step when comparing the model results to observations is to run the modelled clouds through a similar retrieval process. Since the OSIRIS vertical resolution is less than that of the model (1 km as opposed to 0.25 km), the first step is to linearly average the modelled size distributions over four altitude levels. After that the modelled size distributions are passed through the OSIRIS retrieval algorithm, as described in Section 2.1.

In order to investigate how well the retrieval algorithm works, which part of the ice particle size distribution it is sensitive to, and how this is reflected in the retrieved properties, we compare the retrieved modelled clouds to the originally modelled clouds (Figure 4). As the OSIRIS clouds have been retrieved with an assumption of an axial ratio of 2, whereas the modelled clouds are spheres, i.e an axial ratio of 1, we show the retrieved properties for both of these assumptions; axial ratio of 2 in black and axial ratio of 1 in grey. It is clear from this figure that, in general, the two different assumptions generate similar results for the retrieved properties. Indeed, many of the grey markers are hidden by the black markers since the two assumptions give the same results. Panel a shows that the IWC is retrieved rather accurately, for both the "No Wave" (marked with squares) and the "Wave" clouds (marked with stars), even if the retrieved IWC frequently slightly underestimates the volume. This is encouraging since it indicates that IWC is a property we can trust to within approximately 20%.

Panel b shows that the retrieved mean radius generally is larger than the original mean radius by up to about a factor 3 for smaller radii whereas radii of around 50 to 70 nm are well retrieved. The large radii on the other hand (80 nm and above) are greatly underestimated by the retrieval algorithm. The reason is simply that the retrieval algorithm is constrained to select the smaller radii out of two possible solutions, as described in Section 2.1. In practice this prevents the retrieval from retrieving particle sizes above approximately 100 nm. We note that these large radii are mostly produced in the "No Wave" clouds, which, as we shall see in Section 4.4, do not appear to be an adequate representation of the real clouds. Figure 4c shows that small number densities, which generally are associated with fairly large radii at the lower range of the clouds, typically are overestimated, whereas higher number densities are greatly underestimated. The underestimation is worse when an incorrect axial ratio (in this case 2) is assumed (up to a factor 30 for number densities of 1000 particles/cm$^3$) but can still be as large as a factor 10 for the retrieval with the correct axis ratio of 1.





In order to understand the underestimation of high number densities and the overestimation of
small mean radii we study the size distribution. Figure 5 shows a typical example of "Wave"
modelled size distributions at 81 and 84 km respectively (red line), and the retrieved size
distribution using an axial ratio of 2 (black line) and 1 (grey line). Since the retrieval
algorithm assumes a Gaussian distribution it obviously cannot retrieve the bimodal
distributions that often appear in the model. These multi-peaked distributions arise from the
fact that the cold spots produced by atmospheric waves create bursts of newly nucleated
particles. These particles then grow and sediment to a region where older and larger cloud
particles already exist, resulting in a bimodal size distribution. This effect is more prominent
closer to the nucleation region (i.e. the mesopause), and thus the size distribution is more
often multi-peaked at 84 km than at 81 km. Since the retrieval is based on Mie scattering, it is
sensitive mostly to the large end of the particle distribution, and thus will fit a Gaussian to the
larger mode or the larger side of the size distribution. This means that the retrieved mean
radius will be larger than the mean radius of the original size distribution, which explains
what we saw in the middle and bottom panel of Figure 4: For smaller radii (generally higher
in the cloud) the retrieval often overestimates the mean radius, whereas for larger radii around
50 to 70 nm, the agreement is better. Furthermore, the total number densities are generally in
good agreement when number densities are low (typically lower in the cloud where the size
distribution is less bimodal) whereas they are greatly underestimated when number densities
are high (typically higher in the cloud, where the particles in the smaller mode are missed by
the retrieval).
The "No Wave" clouds, which are simulated in a stationary environment lacking the cold
spots that create the bursts of fresh ice particles, generally do not show this behaviour and
thus their size distributions tend to be more Gaussian (see for instance Rapp and Thomas,
2006). In other words a stationary atmosphere typically tends to generate Gaussian size
distributions whereas temperature variations in the atmosphere generate multi-peaked particle
size distributions. This is the reason why the properties of the stationary clouds (squares in
Figure 4) in general are better retrieved and their radii/number densities are not
overestimated/underestimated in the same way as for clouds generated in a non-stationary
atmosphere. It is worth noting that the discussed retrieval issues due to multi-peak
distributions are independent of axial ratio.





**4.4   Comparison to OSIRIS**
We now move on to comparing the raw and retrieved modelled clouds to the OSIRIS
observations. In this section we only show results where an axial ratio of 2 has been assumed
in the retrieval, but the figures look similar and the conclusions remain the same if an axis
ratio of unity is used.
As mentioned earlier the OSIRIS detection threshold as expressed in IWC is approximately 5
$ng/m^3$. In the following we will therefore select only the modelled cloud pixels where the
retrieved ice mass density is higher than this. However, first we investigate how often this is
the case, i.e. the occurrence frequency of clouds above the detection limit. After all, if the
model has an accurate description of the atmospheric state then the occurrence frequency in
the model should be similar to that of the OSIRIS observations. Figure 6 shows the altitude
dependent occurrence rate for the OSIRIS observations (in green), the retrieved "Wave"
clouds (in blue) and the retrieved "No Wave" clouds (in black). The occurrence rate of
"Wave" clouds and the OSIRIS observations both maximise slightly above 20%, and even if
the model suggests that the clouds on average appear 0.5-1 km higher than the observations,
the agreement is still very good. The "No Wave" clouds on the other hand show different
characteristics: the occurrence rate maximise at 80% and the altitude extent of the clouds is
sharply cut off at 81-82 km.
Figure 7 compares profiles of the retrieved properties of the clouds for the "Wave" clouds (in
blue), the "No Wave" clouds (in black) and the OSIRIS clouds (in green). One modelled
profile represents a snapshot of the modelled clouds whereas one OSIRIS profile represents a
retrieved OSIRIS profile. The fat lines represent the mean of all the modelled profiles or the
mean of all OSIRIS profiles. Panel a shows the retrieved radius, panel b the number density
and panel c the IWC. When comparing these properties of the clouds, it is important to
remember that the "No Wave" clouds were tuned to produce the correct IWC by selecting an
appropriate CN distribution, i.e. the black lines of panel c have been tuned so that their
maximum magnitude corresponds to that of the green lines. One should recall that without
this tuning the maximum IWC that developed was only $0.03 \, ng/m^3$. The "Wave" clouds on the
other hand have not been tuned to match the OSIRIS results. Despite the lack of tuning, there





is a good general agreement between the "Wave" clouds and the OSIRIS observations, for all
the three properties; radius, number density, and IWC.
Clearly the "No Wave" clouds are restricted to a more narrow altitude range than the OSIRIS
observations and the "Wave" clouds (the altitudinal range of the "No Wave" clouds is
insensitive to the choice of CN distribution and thus not affected by the aforementioned
tuning). This is easily explained by the static temperature profile, which simply causes
conditions that are too warm for clouds to exist below approximately 82 km (the temperature
reaches 150 K at 82.2 km). In the variable atmosphere on the other hand the clouds can still
exist when the temperature is below the average, which explains the broader altitudinal extent
of the "Wave" clouds and the OSIRIS observations. One may note that below 82 km the
average IWC is higher for the "No Wave" clouds than the OSIRIS clouds, which can be
explained by a difference in temperature variability; the occurrence of cold temperatures
(<150 K) diminishes faster with altitude for OSIRIS than for CMAM (it goes below 50% at
83.3 km for OSIRIS and at 81.8 km for CMAM).
Another aspect where there is better agreement between the OSIRIS observations and the
"Wave" clouds as compared to the "No Wave" clouds is where in the cloud layer the different
quantities peak. For the OSIRIS observations and the "Wave" clouds the number density
generally increases with altitude peaking above the IWC, whereas the mean radii increases
with decreasing altitude and peaks at the bottom of the clouds, i.e. lower than the maximum
IWC. For the "No Wave" clouds the individual profiles for mean radii, number density and
IWC tend to peak at the same altitude (in Figure 7a we can see that some of the individual
profiles show smaller mean radii at 83 km than at 82 km but these are the data points that
were subject to retrieval issues as discussed in Section 4.3 and showed by the squares below
the line in Figure 4b).
To summarize it is clear that the "Wave" clouds agree well with the observations, whereas the
"No Wave" clouds, despite having been tuned to the correct IWC, show different
characteristics.

**5    Conclusions**
In this paper we have used modelled NLC size distributions to investigate the accuracy of the
OSIRIS satellite retrieval algorithm by running it on our modelled distributions and



comparing the retrieved properties to those of the original distributions. We show that IWC is
well retrieved (within 20 %) whereas mean radius and number densities are less accurate. The
retrieved mean radius is often larger than the actual mean radius especially for small radii
where there can be up to a factor of 3 difference. The reason for the inaccuracy is that the
retrieval algorithm assumes a Gaussian size distribution, and when faced with the multimodal
distributions that often occur in the modelled clouds (and thus likely in the real atmosphere),
it will fit a Gaussian to the larger side of the distribution and miss the lower modes, giving an
overestimate of the mean radius. Since the size distributions tend to be more multi-peaked the
closer to the nucleation region one gets, this happens more often higher in the cloud where the
particles are smaller. This explains why the overestimation of the mean radius is more
pronounced for smaller radii. The number density on the other hand, is retrieved fairly well
for small number densities (which generally occur lower in the cloud where the size
distributions are more Gaussian), but is underestimated by a factor of 10 for the high number
densities (which typically occur higher in the clouds where the size distributions are more
multi-peaked).
We proceed to compare the retrieved modelled clouds to those of the OSIRIS tomography
retrieval runs. The temperature and water vapour fields used to drive the model were inferred
from the SMR measurements, which are collocated with the OSIRIS observations of ice
particles. We find that driving the model with stationary temperature and wind fields, as given
by the average of the SMR measurements, does not yield any observable clouds. In fact, for
the model to produce clouds of similar magnitude in ice content as what OSIRIS observes the
average temperature field needs to be reduced by 6 K, and even then the clouds that develop
are not representative for the OSIRIS observations in that they consist of very small number
densities of too large particles. The reason why no clouds develop in the stationary
atmosphere is that the sub-nanometer meteoric smoke particles are too small to be efficient
condensation nuclei at the mesopause temperature of 131 K. We show that by increasing the
size of the CN, and thus making them nucleate more efficiently, it was possible to generate
observable clouds. However, in order to generate clouds of IWC comparable to the OSIRIS
observations, the CN need to be much larger than what we expect from models of transport
and coagulation of meteoric material. Moreover, the characteristics, e.g. the altitudinal extent,
of the clouds produced in this way did not match observations. It is worth pointing out that the
stationary model setup used in Rapp and Thomas (2006) resulted in observable clouds
because they used the meteoric smoke distribution of Hunten et al. (1980) which later have



been shown to greatly overestimate the number of larger (> 1 nm radius) meteoric smoke
particles at the summer mesopause as compared to more advanced models (Megner et al.
2008b, Bardeen et al. 2008).
The region of the atmosphere where NLCs develop is far from stationary, as it is heavily
influenced by wave activity, which infer large fluctuations in the temperature and wind field,
making the actual temperature and winds very different from the average conditions. As a
second step we thus imposed realistic temperature and wind variations on the average SMR
fields and used these varying fields as input for the model. The clouds produced in this way
agree well with OSIRIS observations. Hence, our study suggests that the temperature and
wind variations in the summer mesopause region are what drive the formation of the NLC,
and that the average fields are not enough to quantitatively describe the process of NLC
development. At the same time it is encouraging that a microphysical model, given realistic
varying temperature and wind fields, is capable of producing clouds that, in all by satellite
observable aspects, agree well with the real clouds.
It should be pointed out that there is a clear difference in the size distribution between the
clouds modelled using stationary atmospheric conditions and the more realistic clouds where
varying temperature and wind field have been used. The former often have Gaussian size
distributions whereas the latter most of the time have multimodal size distributions. Since the
latter clouds, in contrast to the former, are in good agreement with observation of the real
clouds, this means that the assumption of a Gaussian (or any single mode) distribution should
be treated with care. While it may still be justified to use a single mode distribution, simply
from the fact that there is a limited number of free parameters one can retrieve using remote
sensing techniques, the user of the data should be cautious of that the number densities and
mean radii retrieved in this way are likely not in agreement with what an in-situ particle
counter would detect.
Finally, we point out that while this study has concentrated on the OSIRIS satellite retrieval
algorithm, the main conclusions should be similar for other satellite retrievals that are based
on scattering techniques and using the same assumptions for retrieving microphysical
parameters.





**Acknowledgements**
The authors would like to thank their colleagues, in particular the people in the particle size
working group, for helpful discussions. Linda Megner was supported by the Swedish
Research Council under contract 621-2012-1648, project 1504401. Victor  I. Fomichev was
supported by the Canadian Space Agency.




**References**

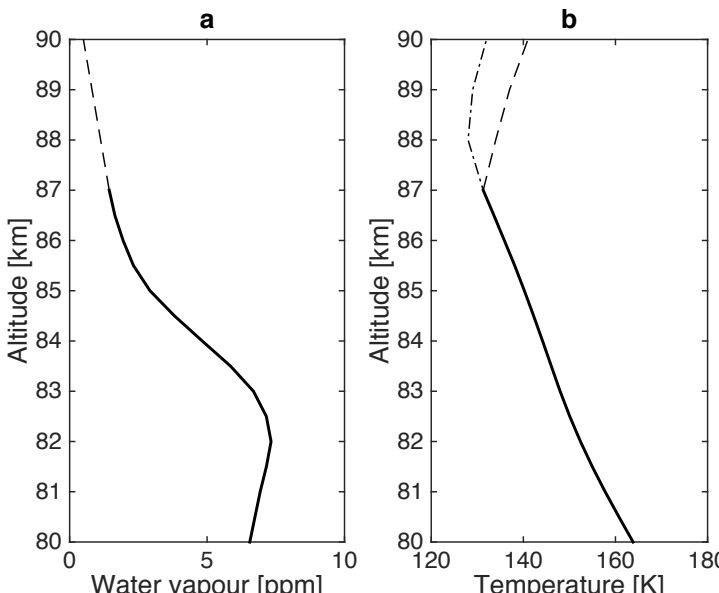

Figure 1. Input data for the "No Wave" model. (a) Average SMR water vapour (solid line)
with a linear extension towards higher altitudes (dashed line). (b) Average SMR temperature
(solid line) extended with SABER data (dashed line) and OSISIS data (dash-dotted line).



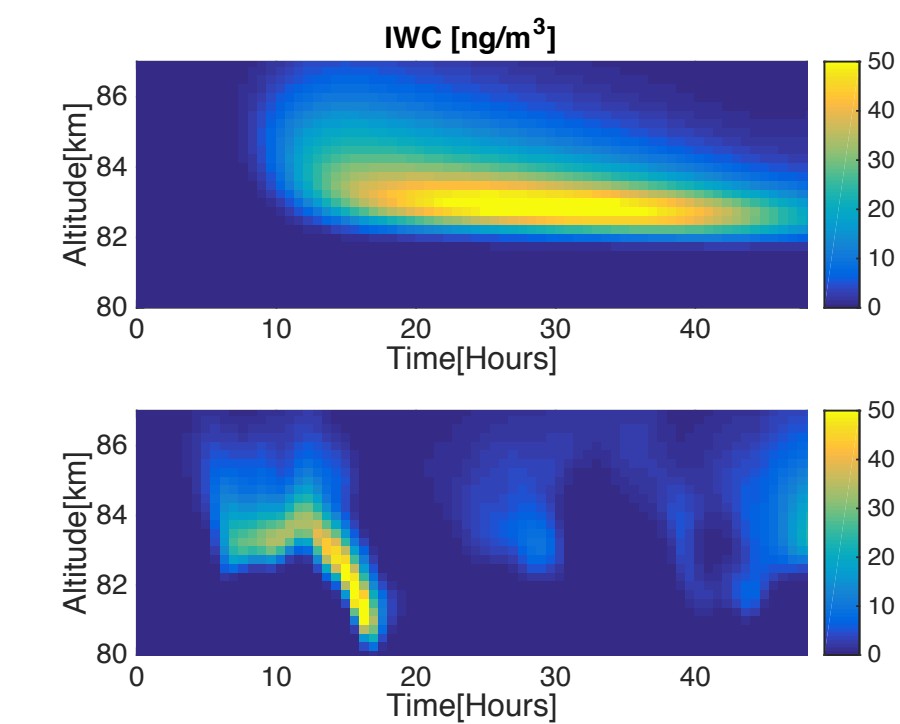

3    Figure 2. IWC of a cloud generated by the "No Wave" model setup (top) and by the "Wave"

4    setup (bottom).





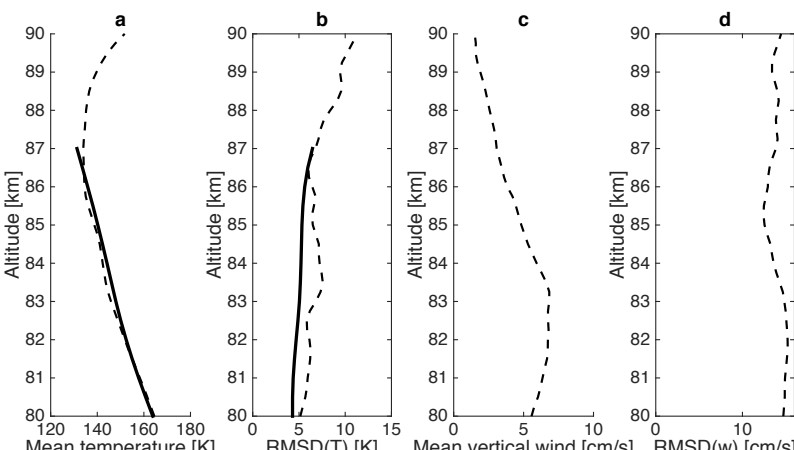

2    Figure 3 Input to the "Wave" model setup (dashed lines). a) Adjusted average CMAM

3    temperature and b) temperature variations. The solid lines show the same quantities for the

4    SMR measurements. c) Average CMAM vertical winds and d) vertical wind variations.



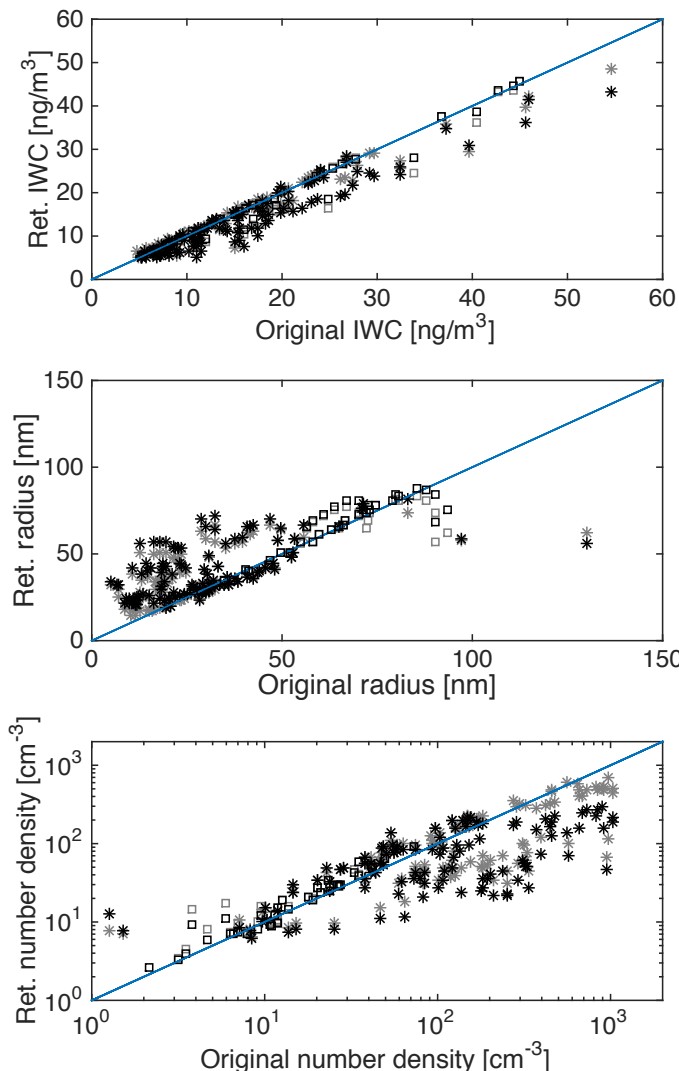

Figure 4 Comparison between properties of the originally modelled clouds and what the
OSIRIS retrieval algorithm calculates. Stars indicate "Wave" clouds and boxes indicate "No
Wave" clouds. Black colour indicates that oblong particles with an axis ratio of 2 were
assumed in the retrieval, and grey colour indicates that spherical particles were assumed.





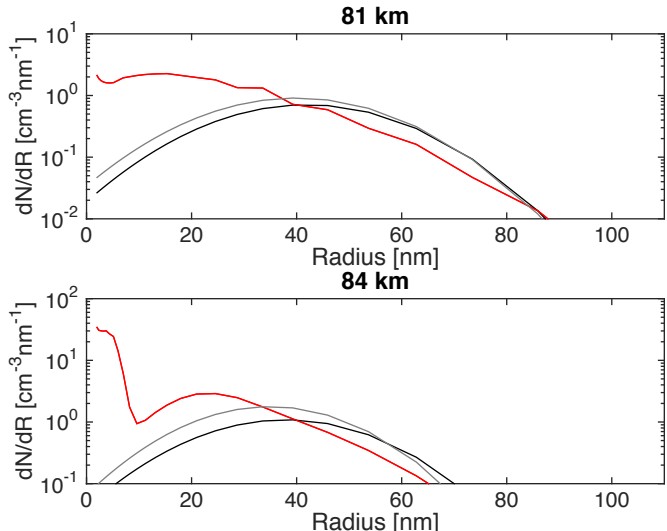

Figure 5. Typical examples of size distributions of the originally modelled clouds (red) and
what is retrieved by OSIRIS using an axial ratio of 2 (black) and of 1 (gray) for an altitude of
81 km (top) and 84 km (bottom).

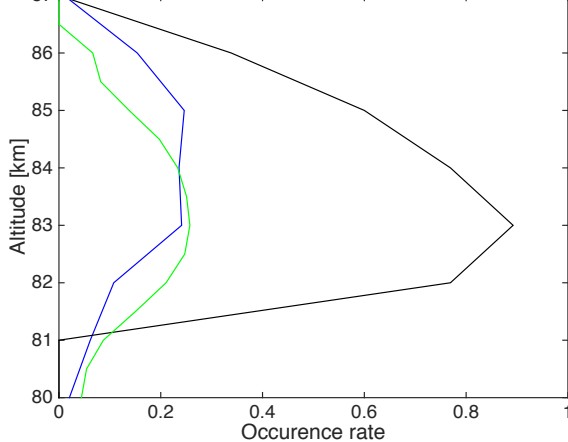

Figure 6. Frequency of occurrence for "Wave" clouds (blue) and "No Wave" clouds (black)
and OSIRIS (green).





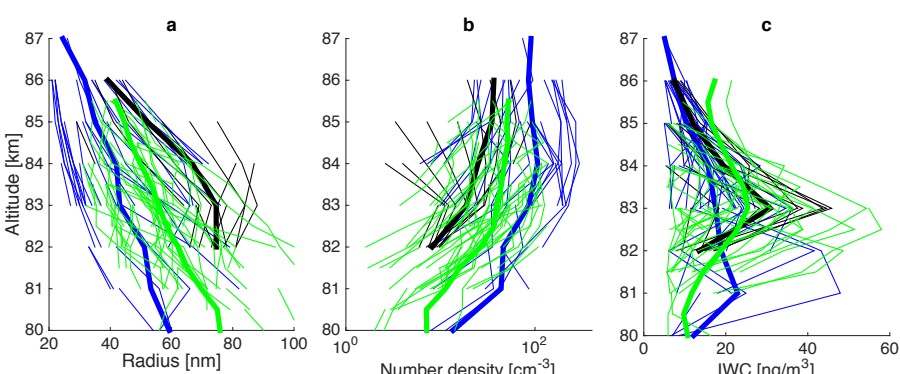

Figure 7. Mean radius profiles for the "Wave" clouds (in blue), the "No Wave" clouds (in black) and the OSIRIS clouds (in green). The modelled profiles represent snapshots of the modelled clouds with 3 h between them for "Wave" clouds and 8 h between for "No Wave" clouds. Only one out of 15 OSIRIS profiles are plotted to avoid a cluttered plot. The fat lines in the figure represent the average of all profiles (not only the plotted ones).



Asmus, H., Robertson, S., Dickson, S., Friedrich, M., and Megner, L.: Charge balance for the mesosphere with meteoric dust particles, Journal of Atmospheric and Solar-Terrestrial Physics, 127, 137-149, 10.1016/j.jastp.2014.07.010, 2015.

Bailey, Thomas, G. E., Hervig, M. E., and Lumpe, J. D.: Comparing nadir and limb observations of polar mesospheric clouds: The effect of the assumed particle size distribution, Journal of Atmospheric Solar-Terrestrial Physics, 127, 51-65, 10.1016/j.jastp.2015.02.007, 2015.

Bardeen, C. G., Toon, O. B., Jensen, E. J., Marsh, D. R., and Harvey, V. L.: Numerical simulations of the three-dimensional distribution of meteoric dust in the mesosphere and upper stratosphere, Journal of Geophysical Research, 113, 10.1029/2007JD009515, 2008.

Barth, C. A., Rusch, D. W., Thomas, R. J., Mount, G. H., Rottman, G. J., Thomas, G. E., Sanders, R. W., and Lawrence, G. M.: Solar Mesosphere Explorer: Scientific objectives and results, Geophysical Research Letters, 10, 237-240, 10.1029/GL010i004p00237, 1983.

Baumgarten, G., Fiedler, J., and Rapp, M.: On microphysical processes of noctilucent clouds (NLC): observations and modeling of mean and width of the particle size-distribution, Atmospheric Chemistry and Physics, 10, 10.5194/acp-10-6661-2010, 2010.

Beagley, S. R., Boone, C. D., Fomichev, V. I., Jin, J. J., Semeniuk, K., McConnell, J. C., and Bernath, P. F.: First multi-year occultation observations of CO2 in the MLT by ACE satellite: observations and analysis using the extended CMAM, Atmos. Chem. Phys., 10, 1133-1153, 10.5194/acp-10-1133-2010, 2010.

Berger, U.: Icy particles in the summer mesopause region: Three-dimensional modeling of their environment and two-dimensional modeling of their transport, Journal of Geophysical Research, 107, 10.1029/2001JA000316, 2002.

Chandran, A., Rusch, D. W., Thomas, G. E., Palo, S. E., Baumgarten, G., Jensen, E. J., and Merkel, A. W.: Atmospheric gravity wave effects on polar mesospheric clouds: A comparison of numerical simulations from CARMA 2D with AIM observations, Journal of Geophysical Research: Atmospheres, 117, 10.1029/2012JD017794, 2012.

Christensen, O. M., Eriksson, P., Urban, J., Murtagh, D., Hultgren, K., and Gumbel, J.: Tomographic retrieval of water vapour and temperature around polar mesospheric clouds using Odin-SMR, Atmospheric Measurement Techniques, 8, 1981-1999, 2015.

Colella, P., and Woodward, P. R.: The Piecewise Parabolic Method (PPM) for gas-dynamical simulations, Journal of Computational Physics, 54, 174-201, http://dx.doi.org/10.1016/0021-9991(84)90143-8, 1984.

Fletcher, N. H.: Size Effect in Heterogeneous Nucleation, The Journal of Chemical Physics, 29, 572-576, doi:http://dx.doi.org/10.1063/1.1744540, 1958.

Fomichev, V. I., Ward, W. E., Beagley, S. R., McLandress, C., McConnell, J. C., McFarlane, N. A., and Shepherd, T. G.: Extended Canadian Middle Atmosphere Model: Zonal-mean climatology and physical parameterizations, Journal of Geophysical Research, 107, 4087, 10.1029/2001JD000479, 2002.




Gumbel, J., and Megner, L.: Charged meteoric smoke as ice nuclei in the mesosphere: Part
1—A review of basic concepts, Journal of Atmospheric and Solar-Terrestrial Physics, 71,
1225-1235, http://dx.doi.org/10.1016/j.jastp.2009.04.012, 2009.
Hansen, G., Serwazi, M., and von Zahn, U.: First detection of a noctilucent cloud by lidar,
Geophysical Research Letters, 16, 1445-1448, 10.1029/GL016i012p01445, 1989.
Hedin, J., Gumbel, J., and Rapp, M.: On the efficiency of rocket-borne particle detection in
the mesosphere, Atmos. Chem. Phys., 7, 3701-3711, 10.5194/acp-7-3701-2007, 2007.
Höffner, J., and Lübken, F. J.: Potassium lidar temperatures and densities in the mesopause
region at Spitsbergen (78°N), Journal of Geophysical Research: Atmospheres (1984–2012),
112, 10.1029/2007jd008612, 2007.
Hultgren, K., Gumbel, J., Degenstein, D., Bourassa, A., Lloyd, N., and Stegman, J.: First
simultaneous retrievals of horizontal and vertical structures of Polar Mesospheric Clouds
from Odin/OSIRIS tomography, Journal of Atmospheric and Solar-Terrestrial Physics, 104,
14  213-223, 2013.

Hultgren, K., and Gumbel, J.: Tomographic and spectral views on the lifecycle of polar
mesospheric clouds from Odin/OSIRIS, Journal of Geophysical Research: Atmospheres, 119,
10.1002/2014jd022435, 2014.
Hunten, D.M., Turco, R.P. and Toon, O.B.: Smoke and dust particles of meteoric origin in the
Mesosphere and Stratosphere, Journal of the Atmospheric Sciences, 37, 1342-1357, 1980
Karlsson, B., and Gumbel, J.: Challenges in the limb retrieval of noctilucent cloud properties
from    Odin/OSIRIS,    Advances    in    Space    Research,    36,    935-942,
http://dx.doi.org/10.1016/j.asr.2005.04.074, 2005.
Keesee, R. G.: Nucleation and particle formation in the upper atmosphere, Journal of
Geophysical Research: Atmospheres, 94, 14683-14692, 10.1029/JD094iD12p14683, 1989.
Leslie, R.: Sky glows, Nature, 32, 245, 1885.
Llewellyn, E. J., Lloyd, N. D., Degenstein, D. A., Gattinger, R. L., Petelina, S., and Bourassa,
A. E.: The OSIRIS instrument on the Odin spacecraft, Canadian Journal of Physics, 82, 411-
422, 10.1139/p04-005, 2004.
Lübken, F.-J., Rapp, M., and Strelnikova, I.: The sensitivity of mesospheric ice layers to
atmospheric background temperatures and water vapor, Advances in Space Research, 40,
794-801, http://dx.doi.org/10.1016/j.asr.2007.01.014, 2007.
Lübken, F. J.: Seasonal variation of turbulent energy dissipation rates at high latitudes as
determined by in situ measurements of neutral density fluctuations, Journal of Geophysical
Research: Atmospheres, 102, 13441-13456, 10.1029/97JD00853, 1997.
McLandress, C., Ward, W. E., Fomichev, V. I., Semeniuk, K., Beagley, S. R., McFarlane, N.
A., and Shepherd, T. G.: Large-scale dynamics of the mesosphere and lower thermosphere:
An analysis using the extended Canadian Middle Atmosphere Model, Journal of Geophysical
Research, 111, 10.1029/2005JD006776, 2006.
Megner, L., Rapp, M., and Gumbel, J.: Distribution of meteoric smoke–sensitivity to
microphysical properties and atmospheric conditions, Atmospheric Chemistry and Physics, 6,
4415-4426, 10.5194/acp-6-4415-2006, 2006.



Megner, L., Gumbel, J., Rapp, M., and Siskind, D. E.: Reduced meteoric smoke particle
density at the summer pole – Implications for mesospheric ice particle nucleation, Advances
in Space Research, 41, 10.1016/j.asr.2007.09.006, 2008a.
Megner, L., Siskind, D. E., Rapp, M., and Gumbel, J.: Global and temporal distribution of
meteoric smoke: A two-dimensional simulation study, Journal of Geophysical Research:
Atmospheres 113, 10.1029/2007JD009054, 2008b.
Megner, L., and Gumbel, J.: Charged meteoric particles as ice nuclei in the mesosphere: Part
2, Journal of Atmospheric and Solar-Terrestrial Physics, 71, 10.1016/j.jastp.2009.05.002,
9  2009.

Megner, L.: Minimal impact of condensation nuclei characteristics on observable
Mesospheric ice properties, Journal of Atmospheric and Solar-Terrestrial Physics, 73,
10.1016/j.jastp.2010.08.006, 2011.
Merkel, A. W., Marsh, D. R., Gettelman, A., and Jensen, E. J.: On the relationship of polar
mesospheric cloud ice water content, particle radius and mesospheric temperature and its use
in multi-dimensional models, Atmos. Chem. Phys., 9, 8889-8901, 10.5194/acp-9-8889-2009,
16 2009.

Nordh, H. L., Schéele, F. v., Frisk, U., Ahola, K., Booth, R. S., Encrenaz, P. J., Hjalmarson,
Å., Kendall, D., Kyrölä, E., Kwok, S., Lecacheux, A., Leppelmeier, G., Llewellyn, E. J.,
Mattila, K., Mégie, G., Murtagh, D., Rougeron, M., and Witt, G.: The Odin orbital
observatory, A&A, 402, L21-L25, 2003.
Rapp, M., and Thomas, G. E.: Modeling the microphysics of mesospheric ice particles:
Assessment of current capabilities and basic sensitivities, Journal of Atmospheric and Solar-
Terrestrial Physics, 68, 10.1016/j.jastp.2005.10.015, 2006.
Roddy, A. F.: The role of meteoric particles in noctilucent clouds, Irish Astronomical Journal,
25 16, 194-202, 1984.

Sheese, P. E., Llewellyn, E. J., Gattinger, R. L., Bourassa, A. E., Degenstein, D. A., Lloyd, N.
D., and McDade, I. C.: Mesopause temperatures during the polar mesospheric cloud season,
Geophysical Research Letters, 38, 10.1029/2011GL047437, 2011.
Stevens, M. H.: The polar mesospheric cloud mass in the Arctic summer, Journal of
Geophysical Research, 110, 10.1029/2004JA010566, 2005.
Toon, O. B., Turco, R. P., Hamill, P., Kiang, C. S., and Whitten, R. C.: A One-Dimensional
Model Describing Aerosol Formation and Evolution in the Stratosphere: II. Sensitivity
Studies and Comparison with Observations, Journal of the Atmospheric Sciences, 36, 718-
736, 10.1175/1520-0469(1979)036<0718:AODMDA>2.0.CO;2, 1979.
Turco, R. P., Hamill, P., Toon, O. B., Whitten, R. C., and Kiang, C. S.: A One-Dimensional
Model Describing Aerosol Formation and Evolution in the Stratosphere: I. Physical Processes
and Mathematical Analogs, Journal of the Atmospheric Sciences, 36, 699-717, 10.1175/1520-
0469(1979)036<0699:AODMDA>2.0.CO;2, 1979.
Turco, R. P., Toon, O. B., Whitten, R. C., Keesee, R. G., and Hollenbach, D.: Noctilucent
clouds: Simulation studies of their genesis, properties and global influences, Planetary and
Space Science, 30, 1147-1181, 1982.
Vergados, P., and Shepherd, M. G.: Retrieving mesospheric water vapour from observations
of volume scattering radiances, Annales Geophysicae, 27, 487-501, 10.5194/angeo-27-487-
44 2009, 2009.




1    Witt, G.: Polarization of light from noctilucent clouds, Journal of Geophysical Research, 65,
2    925-933, 10.1029/JZ065i003p00925, 1960.
