# Peer review of "Comparison of retrieved noctilucent cloud particle"

_Atmospheric Chemistry and Physics, 2016_

## Referee Comment (RC1) · Anonymous Referee #1 · 11 Mar 2016

Review of "Comparison of retrieved Noctilucent cloud particle properties from Odin tomography scans and model simulations" by Megner et al.

**General Comments**

1) The paper investigates the inherent errors in retrieved PMC particle size, concentration, and mass density, when using remote observations. This addressed by using modeled PMC properties to simulate the OSIRIS signals, and then conducting retrievals of size, concentration, and mass density from these signals. Comparisons of the known and retrieved PMC properties give a solid indication of the errors / biases inherent to the observations and the chosen methodology. The conclusions of this paper are important for remote sensing of PMCs. The model based studies indicate that OSIRIS retrievals have greater errors for smaller particle sizes.

2) The second aspect of the paper is to determine if inclusion of atmospheric waves in PMC microphysical models gives a better reproduction observed PMC properties, compared to using a static atmosphere. The Author's find that simulations with waves indeed give the best explanation of observed PMC properties, as shown in Figures 6 and 7. The conclusions here are important for PMC modeling efforts, however, the representation of waviness in the model (section 4.2) is somewhat brief. Is it possible to describe the wave parameters used in more detail, perhaps in such a way that other modelers could implement a scheme like yours? Also, the agreement is Fig 7 between the OSIRIS and wavy model is not spot on. Is it possible that some wave tuning would give better agreement (and thus indicate a refined picture of the relevant waves)?

**Specific Comments**

Throughout: "modelled" should be "modeled".

Throughout: In PMC / NLC literature "IWC" usually refers to the vertically integrated water content (g / km$^2$). You assign IWC units of ng/m$^3$, which would be ice mass density ($m_i$). You need to change IWC for $m_i$ (or $M_i$) throughout. (I know IWC is a clumsy and probably misplaced acronym, but it is widely recognized as g / km$^2$ in the PMC field).

p 1 line 12: I don't think we capitalize Noctilucent, or Polar Mesospheric Cloud.

p 1 line 13: add "ground based remote sensing" to the list

p 1 line 19: "…on signals based on modeled…"

p 2 line 10: This statement is missing something, you state that PMCs are a means to monitor the atmosphere, but do not state which aspects of the atmosphere.

p 2 line 25: "number density" typically refers to the number of gas molecules per cc. If you are referring to ice particles, then typical nomenclature would be "ice concentration (N)".

p 3 lines 4-7: The comment in parenthesis can just be a sentence.

p 3 line 12: Again, to be precise, you retrieve PMC properties from signals simulated using modeled size distributions.

p 4 line 18: "…spectral resolution of…"

p 4 line 25: I think this should be "…fixed at 16 nm for radii larger than 40 nm…". You should also state that many other remote sensing PMC experiments have adopted this assumption, e.g. CIPS, SOFIE, SCHIAMACHY, SBUV, & probably others.

p 4 line 27: Is there a reference that supports the choice of AR = 2?

p 4 lines 27-30: You are describing the two-valued solutions for certain conditions. This could be stated more clearly.

p 5 line 15: It would be useful to state the SMR vertical and horizontal resolution.

p 6 line 13: Here you should cite the recent study by Killani et al (ACP 2015) that deals with non-spherical ice in microphysical PMC models. The main point is that there are microphysical effects due to non-spherical shapes that change the modeled PMC properties, in addition to the well known optical effects of non-spherical ice.

p 6 line 29: You should also mention that ice sublimation enhances vapor at the ice layer bottom. Does SMR detect the dry and wet regions associated with ice?

p 7 line 25: "(fraction of 1 nm)" should be stated as "(radii < ~1 nm)"

Figure 1: You should add the frost point temperature vs. height, this would make your arguments on p 7 flow very easily. Also, it would be instructive to add error bars as the standard deviations to give an idea of the natural variability. "OSISIS" - "OSIRIS"

p 8, lines 17-18 & 32 (and elsewhere): You often mix units and nomenclature for ice mass. For example "ice water density" is stated as being in $ng/m^3$, where I would consider these units to be associated with "ice mass density". Later you refer to "ice mass" which I assume is "ice mass density". Perhaps introduce a variable "m sub i" if that would make the discussion more convenient, in any case make the language consistent.

p 9 line 1: What specifically is the OSIRIS IWC observation mentioned here? Is it the average associated with the SMR data in Figure 1, or something else?

p 9 line 25: By "constant" do you mean "constant in height" ?

p 10 line 11: "less than" should be "broader than"

p 10 line 13: Do you really pass the model size distributions through the OSIRIS retrieval algorithm? I would think that you use the model distributions to simulate OSIRIS signals, and then pass these signals through the retrieval code. Please clarify.

p 10 line 19: I think you mean that the microphysical treatment of ice particles in CARMA assumes spheres. But when you do the OSIRIS signal simulations, do you assume spheres or AR=2? This aspect of the signal simulation should be stated. Again, Killani et al. [2015] discuss the microphysical implications AND the optical retrieval implications for non-spherical NLC particles, and that work is relevant to your study and thus should be mentioned.

p 10 line 28: Please clarify what "mean radius" refers to (e.g., numeric mean, mass weighted mean, the Gaussian median, …).

p 10 line 28: Panel b of which figure?

p 11 line 7-9: Part of the challenge is that the error in concentration (N) is proportional to the cube of the radii error. The propagation of radii errors into the other values exists because you determine radii first, and then mass density and N ( presumably based on the modeled signal based on retrieved radii).  In any case, you should discuss further the reasons for N having the greatest errors.

p 11 line 20: The retrieval cannot be based on Mie scattering since you accommodate non-spherical particles. Indeed, you state above that the optical calculations are from the T-matrix algorithm.

p 11 lines 21-22: There may be a better explanation for why the retrieval indicates larger particles than the numeric mean. I suspect the reason is that the smallest particles do not contribute to the OSIRIS signal. I think this would be evident if you plot the fraction of total radiance in each size bin of the size distribution. If this explains the discrepancy (I think it will), then showing the additional figure would be very useful (I don't think anyone has published this and it could settle some old debates).

p 12 lines 18-28, and Figure 6: You switch between "rate" and "frequency", the later would be convention.

Figure 7: This might be clearer if you showed standard deviations instead of all the individual profiles (thin lines).

p 13 line 20: The statement "…exist when the temperature is below the average,…" is unclear. The average is of what group of data?

p 13 lines 21-25: The no wave case (thick black) is zero below 82 km, so the statement does not make sense. Perhaps you meant the wave case. You should remind us to look at Figure 7c.

p 13 lines 28-32: Some of this is hard to see because of the many thin lines in the plots. I do, however, see your basic points here, and you should not that both the ALOMAR lidar and SOFIE have shown this behavior as well, where N peaks at an altitude above the peak in ice mass density, and radii are largest below the peak in mass density.

p 16 line 7: I believe the correct name is "PMC microphysics and happy hour working group".

---

## Short Comment (SC1) · 4 May 2016

Dear authors,

I found it very interesting to read your paper, in particular the results regarding the influence of background variability on NLC properties.

For improving the soundness of your conclusions, I have a suggestion regarding your CARMA simulations. While conducting CARMA simulations with gravity wave perturbed background fields, I found that different dynamical situations cause very different NLC properties, including very different size distributions. It seems that in your wave-simulation only one bright NLC can develop (probably due to the limited water vapor).

[Figure]

Therefore, it might be insightful to start the simulation also at later times. This could give you a larger variety of NLC properties and increase the number of datapoints in Fig. 4, in particular more datapoints for large ice water content.

The importance of dynamical fluctuations is nicely highlighted in your analysis. Similar results and a detailed discussion about the factors limiting the microphysics in gravity wave perturbed backgrounds can be found in our recent publication (Wilms et al., Nucleation of mesospheric cloud particles: Sensitivities and limits, JGR, 2016, doi: 10.1002/2015JA021764). In order to compare your result to the above mentioned study, it would be helpful to see the background temperature and wind fields which were used in your simulations, in addition to the mean properties and variances shown in Fig. 3. As one of your main conclusions refers to the importance of the variability of the background atmosphere, I believe it is important to characterize this variability more closely (e.g., dominant periods, wavelengths, phase relation between temperature and vertical wind).

I am looking forward to your new results.

Kind regards, Henrike Wilms

---

## Referee Comment (RC2) · Anonymous Referee #2 · 27 May 2016

The authors have responded to my initial comments very well. The one change I would suggest from the original comments is that the authors should point out (as they did in the response to the reviewer) in the paper that they were able to reproduce the Rapp and Thomas work to ensure their use of the CARMA model is consistent with other users.

Since my original review, a comment has been submitted from H. Wilms. I agree with his suggestion that adding a figure showing background winds and temperature would facilitate comparisons to other works. I don't see that as essential, but concur that it would be an improvement. I suggest the authors consider it, but I would not require it.

Otherwise, I believe this paper is acceptable as is.

---

## Author Comment (AC1) · 15 Sep 2016

We thank both reviewers and Henrike Wilms for their many helpful comments, most of which we have incorporated in the new version of our manuscript (found below with changes in blue).

In the time since the first submission of this manuscript the OSIRIS data was updated.  We have also implemented the suggestion of Henrike Wilms of starting nucleation also at later times, and in order for all simulations to be of the same length, we have restricted the simulation time to 24 hours.  This gave us a more varied data set and has resulted in some smaller changes in the results, as can be seen in some of the figures, but the main messages remain. For instance, the occurrence rate in figure 8 has changed somewhat. The reason that, since it takes some time for the clouds to develop, the occurrence rate is somewhat sensitive to the arbitrary chosen start and end time of the simulation, and thus changed as we use 24 h instead of 48 h simulation time. Nevertheless, the occurrence frequency for the Wave Clouds remain in better agreement with OSIRIS than the No Wave clouds. We have updated the text to comply with these changes.

We have also changed the y-scale in figure 7 from logarithmic to linear in order to make it more clear how insensitive the particle retrieval is to the smaller sizes.

Reply to reviewer 1 (The comments of the reviewer are in *italic*):

*General Comments*

*1) The paper investigates the inherent errors in retrieved PMC particle size,*

*concentration, and mass density, when using remote observations. This addressed by*

*using modeled PMC properties to simulate the OSIRIS signals, and then conducting*

*retrievals of size, concentration, and mass density from these signals. Comparisons of the*

*known and retrieved PMC properties give a solid indication of the errors / biases inherent to*

*the observations and the chosen methodology. The conclusions of this paper are important for*

*remote sensing of PMCs. The model based studies indicate that OSIRIS retrievals have greater*

*errors for smaller particle sizes.*

*2) The second aspect of the paper is to determine if inclusion of atmospheric waves in*

*PMC microphysical models gives a better reproduction observed PMC properties,*

*compared to using a static atmosphere. The Author's find that simulations with waves*

*indeed give the best explanation of observed PMC properties, as shown in Figures 6 and 7. The*

*conclusions here are important for PMC modeling efforts, however, the*

*representation of waviness in the model (section 4.2) is somewhat brief. Is it possible to describe*

*the wave parameters used in more detail, perhaps in such a way that other*

*modelers could implement a scheme like yours? Also, the agreement is Fig 7 between the*

*OSIRIS and wavy model is not spot on. Is it possible that some wave tuning would give better*

*agreement (and thus indicate a refined picture of the relevant waves)?*

In response to this comment and the one from Henrike Wilms we have added a figure with the temperature and wind fields used so that other modellers can compare to their work, or use similar fields if they wish.

The idea of tuning the model to give a better representation of the waves is indeed interesting.

The current data set is too small to allow for this but it may be possible with the upcoming

MATS satellite which will use tomographic retrieval to get 3D information of temperature and

PMC and where a much larger data set will be available. We have added a discussion of this idea to the manuscript.

*Specific Comments*

*Throughout: "modelled" should be "modeled".*

According to the windows spelling checker "modelled" is UK English and "modeled" US

English. Since ACP is a European journal we assume UK spelling should be used and thus leave this as it is.

*Throughout: In PMC / NLC literature "IWC" usually refers to the vertically integrated water*

*content (g / km$_2$). You assign IWC units of ng/m$_3$, which would be ice mass density ($m_i$). You*

*need to change IWC for $m_i$ (or $M_i$) throughout. (I know IWC is a clumsy and probably misplaced*

*acronym, but it is widely recognized as $g / km_2$ in the PMC field).*

This has been changed according to the suggestion.

*p 1 line 12: I don't think we capitalize Noctilucent, or Polar Mesospheric Cloud.*

This has been changed according to the suggestion.

*p 1 line 13: add "ground based remote sensing" to the list*

This has been changed according to the suggestion.

*p 1 line 19: "...on signals based on modeled..."*

This has been added according to the suggestion.

*p 2 line 10: This statement is missing something, you state that PMCs are a means to*

*monitor the atmosphere, but do not state which aspects of the atmosphere.*

This has been changed to "…a way to monitor changes in this remote region of the atmosphere…"

*p 2 line 25: "number density" typically refers to the number of gas molecules per cc. If you are*

*referring to ice particles, then typical nomenclature would be "ice concentration (N)".*

This has been changed according to the suggestion.

*p 3 lines 4-7: The comment in parenthesis can just be a sentence.*

This has been changed according to the suggestion.

*p 3 line 12: Again, to be precise, you retrieve PMC properties from signals simulated*

*using modeled size distributions.*

This has been changed according to the suggestion.

*p 4 line 18: "...spectral resolution of..."*

This has been changed according to the suggestion.

*p 4 line 25: I think this should be "...fixed at 16 nm for radii larger than 40 nm...". You should*

*also state that many other remote sensing PMC experiments have adopted this assumption, e.g.*

*CIPS, SOFIE, SCHIAMACHY, SBUV, & probably others.*

This has been changed according to the suggestion.

*p 4 line 27: Is there a reference that supports the choice of AR = 2?*

Reference added.

*p 4 lines 27-30: You are describing the two-valued solutions for certain conditions. This could*
*be stated more clearly.*

This has been changed according to the suggestion.

*p 5 line 15: It would be useful to state the SMR vertical and horizontal resolution.*

This has been added.

*p 6 line 13: Here you should cite the recent study by Killani et al (ACP 2015) that deals with*
*non-spherical ice in microphysical PMC models. The main point is that there are microphysical*
*effects due to non-spherical shapes that change the modeled PMC properties, in addition to the*
*well known optical effects of non-spherical ice.*

This has been added according to the suggestion.

*p 6 line 29: You should also mention that ice sublimation enhances vapor at the ice layer*
*bottom. Does SMR detect the dry and wet regions associated with ice?*

This has been mentioned. Yes, the SMR data show several regions of enhanced water vapour
concentration at the altitudes corresponding to the lower layer of NLCs. However, as the water
vapour enhancement created by a cloud can linger far after a cloud has sublimated, a direct
correlation between cloud presence and enhancement is not seen in the dataset.

*p 7 line 25: "(fraction of 1 nm)" should be stated as "(radii < ~1 nm)"*

This has been changed according to the suggestion.

*Figure 1: You should add the frost point temperature vs. height, this would make your*
*arguments on p 7 flow very easily. Also, it would be instructive to add error bars as the standard*
*deviations to give an idea of the natural variability. "OSISIS" - "OSIRIS"*

The standard deviation of the OSIRIS temperature is shown in figure 5. Here we choose to only
present input to the stationary model.

*p 8, lines 17-18 & 32 (and elsewhere): You often mix units and nomenclature for ice*
*mass. For example "ice water density" is stated as being in ng/m$_3$, where I would*
*consider these units to be associated with "ice mass density". Later you refer to "ice*

*mass" which I assume is "ice mass density". Perhaps introduce a variable "m sub i" if*

*that would make the discussion more convenient, in any case make the language*

*consistent.*

We now use "ice mass density" or "$m_i$" troughout the manuscript.

*p 9 line 1: What specifically is the OSIRIS IWC observation mentioned here? Is it the*

*average associated with the SMR data in Figure 1, or something else?*

This has been clarified.

*p 9 line 25: By "constant" do you mean "constant in height" ?*

We do not find the wording "constant" at the indicated place but we have clarified what we mean by "constant" in 4.2.

*p 10 line 11: "less than" should be "broader than"*

There appears to be something wrong with the page and line indications. We assume the reviewer means the vertical resolution of OSIRIS and has changed that according to the suggestion.

*p 10 line 13: Do you really pass the model size distributions through the OSIRIS retrieval*
*algorithm? I would think that you use the model distributions to simulate OSIRIS signals, and*
*then pass these signals through the retrieval code. Please clarify.*

This has been reformulated.

*p 10 line 19: I think you mean that the microphysical treatment of ice particles in*

*CARMA assumes spheres. But when you do the OSIRIS signal simulations, do you*

*assume spheres or AR=2? This aspect of the signal simulation should be stated. Again, Killani*
*et al. [2015] discuss the microphysical implications AND the optical retrieval implications for*
*non-spherical NLC particles, and that work is relevant to your study and thus should be*
*mentioned.*

Yes, that is what we meant, this had been clarified and the work of Kiliani et al. is mentioned.

*p 10 line 28: Please clarify what "mean radius" refers to (e.g., numeric mean, mass*

*weighted mean, the Gaussian median, ...).*

This has been added to the figure caption.

*p 10 line 28: Panel b of which figure?3*

Yes, this has been corrected.

*p 11 line 7-9: Part of the challenge is that the error in concentration (N) is proportional to the*
*cube of the radii error. The propagation of radii errors into the other values exists because you*
*determine radii first, and then mass density and N ( presumably based on the modeled signal*
*based on retrieved radii). In any case, you should discuss further the reasons for N having the*
*greatest errors.*

This has been added.

*p 11 line 20: The retrieval cannot be based on Mie scattering since you accommodate*

*non-spherical particles. Indeed, you state above that the optical calculations are from the T-*
*matrix algorithm.*

This has been reformulated.

*p 11 lines 21-22: There may be a better explanation for why the retrieval indicates larger*
*particles than the numeric mean. I suspect the reason is that the smallest particles do not*
*contribute to the OSIRIS signal. I think this would be evident if you plot the fraction of total*
*radiance in each size bin of the size distribution. If this explains the discrepancy (I think it will),*
*then showing the additional figure would be very useful (I don't think anyone has published this*
*and it could settle some old debates).*

Yes, this is exactly what we mean with that OSIRIS is more sensitive to larger particles – so
that smallest particles will not contribute to the signal.  This is the reason we do not see the
many small particles. But if we have a multimodal distribution with one mode at a very small
particle size then the effect is that we miss that mode completely and thus make a large error in
the estimate of total ice particle concentration. We have changed the text so that this becomes
clearer and we have changed the y-scale of figure 7 so that it becomes even more obvious how
much of the particles we are insensitive too. We liked the idea of such a plot and made it (see
below, note that the y-axis differs slightly from the one in the paper) but we found that it is
confusing for the reader. Therefore, we simply shaded the region where 90% of the signal
comes from in the paper instead.

[Figure]

*p 12 lines 18-28, and Figure 6: You switch between "rate" and "frequency", the later*

*would be convention.*

This has been changed.

*Figure 7: This might be clearer if you showed standard deviations instead of all the*

*individual profiles (thin lines).*

We have changed this plot in line with the suggestion apart from that we use percentiles since
the distributions are non-normal.

*p 13 line 20: The statement "...exist when the temperature is below the average,..." is*

*unclear. The average is of what group of data?*

This has been reformulated.

*p 13 lines 21-25: The no wave case (thick black) is zero below 82 km, so the statement does not make sense. Perhaps you meant the wave case. You should remind us to look at Figure 7c.*

Yes, we meant the wave case. Thank you.

*p 13 lines 28-32: Some of this is hard to see because of the many thin lines in the plots. I do, however, see your basic points here, and you should not that both the ALOMAR lidar and SOFIE have shown this behavior as well, where N peaks at an altitude above the peak in ice mass density, and radii are largest below the peak in mass density.*

This has been removed. It still seems to be the case if you look at individual profiles but if you look at the mean, median or percentiles it is not the case. (These properties becomes different from individual profiles because at altitudes that clouds are the most frequent they are the calculated from many profiles but as we go to altitudes where the clouds rarer there are less profiles left from which these properties are calculated and those that are left are those of the brighter clouds.) So we have chosen to remove this part.

*p 16 line 7: I believe the correct name is "PMC microphysics and happy hour working group".*

Indeed. We thank the reviewer, who clearly is an expert in the field, but keep the current wording in order to not confuse more the general public (smile).

Reply to reviewer 2:

We agree that we should point out more clearly that we can reproduce the results of Rapp and Thomas and have done so in the updated manuscript. We have also added a figure with the temperature and wind fields used as input for the model.

Reply to Henrike Wilms:

We have added a figure showing the temperature and wind fields used as input for our model. We have also now started the nucleation process at not only 0 h, 10 h, and 20h do get more variations in the clouds.

[revised manuscript text omitted]